# SCOUT: Teaching Pre-trained Language Models to Enhance Reasoning via Flow Chain-of-Thought

**Guanghao Li**[1,2,*]   **Wenhao Jiang**[3,*]   **Mingfeng Chen**[1]   **Yan Li**[4]   **Hao Yu**[1]

**Shuting Dong**[1]   **Tao Ren**[5]   **Ming Tang**[2,†]   **Chun Yuan**[1,†]

[1]Tsinghua Shenzhen International Graduate School, Tsinghua University
[2]Southern University of Science and Technology
[3]Guangdong Laboratory of AI and Digital Economy (SZ)
[4]The Hong Kong University of Science and Technology
[5]Guanghua School of Management, Peking University
ligh24@mails.tsinghua.edu.cn  cswhjiang@gmail.com
yuanc@sz.tsinghua.edu.cn  tangm3@sustech.edu.cn

## Abstract

Chain-of-Thought (CoT) prompting improves the reasoning performance of large language models (LLMs) by encouraging step-by-step thinking. However, CoT-based methods depend on intermediate reasoning steps, which limits scalability and generalization. Recent work explores recursive reasoning, where LLMs reuse internal layers across iterations to refine latent representations without explicit CoT supervision. While promising, these approaches often require costly pretraining and lack a principled framework for how reasoning should evolve across iterations. We address this gap by introducing **Flow Chain-of-Thought (Flow CoT)**, a reasoning paradigm that models recursive inference as a progressive trajectory of latent cognitive states. Flow CoT frames each iteration as a distinct cognitive stage—deepening reasoning across iterations without relying on manual supervision. To realize this, we propose **SCOUT** (*Stepwise Cognitive Optimization Using Teachers*), a lightweight fine-tuning framework that enables Flow CoT-style reasoning without the need for pretraining. SCOUT uses progressive distillation to align each iteration with a teacher of appropriate capacity, and a cross-attention-based retrospective module that integrates outputs from previous iterations while preserving the model's original computation flow. Experiments across eight reasoning benchmarks show that SCOUT consistently improves both accuracy and explanation quality, achieving up to 1.8% gains under fine-tuning. Qualitative analyses further reveal that SCOUT enables progressively deeper reasoning across iterations—refining both belief formation and explanation granularity. These results not only validate the effectiveness of SCOUT, but also demonstrate the practical viability of Flow CoT as a scalable framework for enhancing reasoning in LLMs.

## 1   Introduction

Chain-of-Thought (CoT) reasoning improves large language models' (LLMs') performance by prompting them to generate intermediate steps that mimic human thought processes [1, 2, 3]. Fundamentally, CoT trades additional computation for improved accuracy. However, most CoT-based methods rely on supervised learning over curated step-by-step data, which limits their generalizability

---

[*]Equal contribution
[†]Corresponding author

39th Conference on Neural Information Processing Systems (NeurIPS 2025).

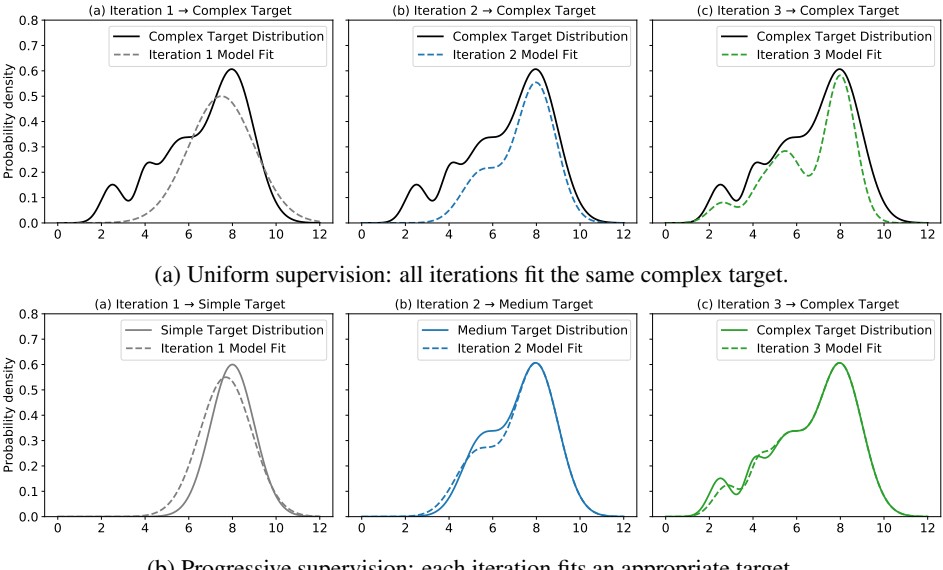

(a) Uniform supervision: all iterations fit the same complex target.

(b) Progressive supervision: each iteration fits an appropriate target.

Figure 1: Motivation comparison between progressive and uniform supervision training. Progressive supervision aligns the supervision signal with the model's evolving capacity, avoiding over-regularization in early stages and undertraining in later ones.

and scalability [4]. To address these limitations, recent studies have explored *recursive reasoning*, in which LLMs are iteratively applied to refine internal representations in latent space [5]. While these approaches increase computational depth without requiring explicit CoT supervision, they often rely on costly pretraining and still lack a principled framework for how reasoning should evolve across iterations. In this work, we extend this line of research and introduce **Flow Chain-of-Thought (Flow CoT)**, a reasoning paradigm that conceptualizes recursive inference as a progressive trajectory of cognitive states. Each recursive iteration in Flow CoT represents a refined reasoning stage, mirroring how human cognition gradually unfolds. Crucially, Flow CoT enables reasoning to emerge from recursive computation—without reliance on labeled CoT traces.

Although Flow CoT provides a compelling reasoning paradigm, existing recursive methods face two main obstacles in applying it effectively. First, they supervise all iterations using identical hard labels, ignoring that each iteration differs in capacity and function. Each iteration serves a dual role: generating a plausible output and constructing an internal representation that supports future reasoning [6]. Applying strong supervision to early iterations can over-regularize them and distort intermediate states, thereby impairing downstream iterations. As illustrated in Figure 1a, enforcing a single target distribution across all iterations is suboptimal for capturing the inherently progressive nature of reasoning. Second, most recursive methods implement simple strategies like feeding previous outputs back as inputs [7, 8] or concatenating them with the original prompt [4]. While these mechanisms are often used during pretraining, their reliance on modifying the model's input-output structure makes them challenging to integrate into fine-tuning pipelines.

To overcome these challenges, we propose **SCOUT** (*Stepwise Cognitive Optimization Using Teachers*), a lightweight fine-tuning framework that equips pretrained LLMs with Flow CoT capabilities. SCOUT introduces a *progressive distillation strategy*, where each reasoning step is supervised by a teacher model of matching strength. As illustrated in Figure 1b, this enables each iteration to refine its reasoning based on current capacity, rather than being constrained by uniform, overly strong supervision targets. In addition, we incorporate a non-intrusive *retrospective module* based on cross-attention, which allows each step to selectively attend to previous outputs while preserving the model's original reasoning flow. Together, these components preserve the base model's capabilities while significantly enhancing reasoning through fine-tuning alone. SCOUT thus serves as a practical realization of the Flow CoT paradigm, offering a structured and cognitively grounded alternative to conventional recursive reasoning. Experiments across diverse benchmarks confirm consistent improvements in both accuracy and explanation quality, demonstrating the scalability and effectiveness of Flow CoT through fine-tuning alone.

We summarize our main contributions as follows:

- We propose Flow Chain-of-Thought (Flow CoT), a new paradigm that conceptualizes recursive reasoning as a progressive, depth-aware cognitive trajectory.
- We introduce SCOUT, a fine-tuning framework that instills Flow CoT capabilities in pretrained LLMs via progressive distillation with multiple teacher models and a lightweight retrospective module, enabling recursive reasoning without additional pretraining.
- We conduct extensive experiments across eight reasoning benchmarks. SCOUT not only achieves up to 1.8% accuracy gains under fine-tuning, but also consistently improves reasoning coherence and explanation quality.

## 2  Related Work

**Chain-of-Thought Reasoning.** Large language models (LLMs) have demonstrated remarkable capabilities across a wide range of tasks [9, 10, 11]. Chain-of-Thought (CoT) methods elicit intermediate reasoning steps in natural language to enhance the performance of LLMs. Early work demonstrates that prompting LLMs with exemplar reasoning chains can significantly improve their accuracy on complex tasks [1, 12, 13]. Subsequent approaches fine-tune models to generate reasoning steps via supervised learning [14, 15] or reinforcement learning [16, 17, 18]. These methods increase the model's effective reasoning depth by recursively incorporating generated thoughts into the input context [19]. Recent advances focus on improving efficiency, including generating concise reasoning traces [20], deferring decisions until sufficient computation has occurred [21], or performing reasoning entirely in latent space using continuous representations [22, 23]. However, most CoT methods require explicitly annotated intermediate steps, limiting scalability and generalizability across domains.

**Recursive LLMs models.** Recurrent computation has been explored in various transformer designs, such as Universal Transformers [24], Looped Transformers [25, 26], and Sparse Recurrent Transformers [27]. These ideas have recently been adapted to LLMs to support multi-iteration reasoning via recursive self-application [28, 8, 5, 29]. Recursive refinement deepens the reasoning trajectory by reprocessing the model's own outputs [7, 4]. However, most recursive methods rely on simple heuristics such as feeding previous outputs back as inputs [7, 8] or concatenating them with the original prompt [4]. While effective during pretraining, these strategies are often difficult to adapt for fine-tuning without modifying model architecture or training pipelines. In addition, these methods typically apply uniform supervision across all iterations, ignoring the evolving representational capacity at different depths and risking misaligned learning signals, particularly in early-stage reasoning.

**Knowledge Distillation.** Knowledge distillation transfers information from a stronger teacher model to a smaller student, commonly to compress models while retaining performance [30, 31, 32]. Standard approaches include sequence-level distillation (SeqKD) [33] and distribution matching via Kullback-Leibler divergence [34, 35, 36]. Others align internal representations directly [37]. However, overly strong teachers can destabilize training, especially when student capacity is limited [38, 39, 40].

**Positioning Our Work.** SCOUT provides a practical realization of Flow CoT by enabling multi-iteration reasoning without requiring explicitly annotated intermediate steps. Unlike prior recursive LLMs that apply uniform supervision or require extensive pretraining, SCOUT employs progressive distillation—aligning each iteration with a teacher of matching capacity—and introduces a lightweight retrospective module. This allows pretrained LLMs to acquire depth-aware reasoning capabilities directly through fine-tuning.

## 3  Method

### 3.1  Flow Chain-of-Thought

Many recent reasoning methods for large language models (LLMs) rely on recursive self-application, yet they typically treat each iteration as a black-box repetition without modeling the evolving nature of cognition. To address this, we introduce **Flow Chain-of-Thought (Flow CoT)**, a general framework that models multi-step reasoning as a latent trajectory of progressively refined cognitive states.

Inspired by recent iterative reasoning frameworks [4, 6], we adopt a standard three-part decomposition—which we extend and reinterpret to support Flow CoT's stepwise cognitive model-

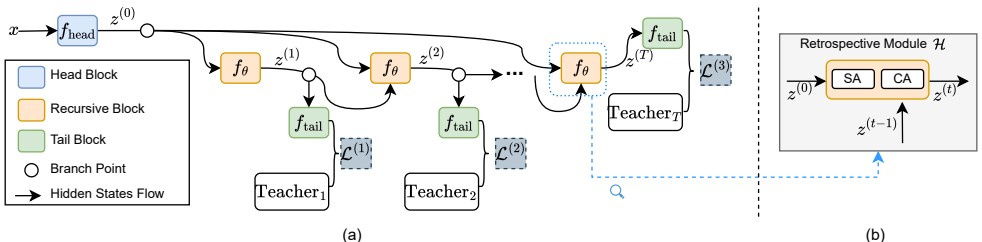

Figure 2: **SCOUT Architecture.** (a) Overall pipeline: a pretrained LLM is decomposed into a head block $f_{\text{head}}$, a recursive block $f_\theta$, and a tail block $f_{\text{tail}}$. The model performs $T$ reasoning steps, each producing a latent state $z^{(t)}$, which is decoded by tail block and supervised by a capacity-matched teacher. Teacher strength increases with $t$ (i.e., $\text{Teacher}_t > \text{Teacher}_{t-1}$), enabling progressive learning. (b) The retrospective module integrates $z^{(t-1)}$ via cross-attention (CA), while self-attention (SA) attends to the initial state $z^{(0)}$, enhancing step-to-step coherence.

ing—consisting of a *head block* $f_{\text{head}}$, a *recursive block* $f_\theta$, and a *tail block* $f_{\text{tail}}$. This design enables the head block to encode the input once, the recursive block to iteratively refine internal representations, and the tail block to decode the final output.

Given a discrete input prompt $x$, the head block produces an initial latent state $z^{(0)} = f_{\text{head}}(x)$, which is then refined through $T$ recursive steps. The first iteration applies the recursive block directly to the initial state:

$$z^{(1)} = f_\theta(z^{(0)}), \tag{1}$$

while subsequent iterations integrate both the original context and intermediate reasoning traces:

$$z^{(t)} = f_\theta\left(\mathcal{H}(z^{(0)}, z^{(t-1)})\right), \quad t = 2, \ldots, T. \tag{2}$$

Here, $\mathcal{H}$ is a history integration function that merges the fixed input context with the evolving internal state. This formulation reflects a key assumption of Flow CoT: effective reasoning requires both retention of prior progress and grounding in the original question. The integration of $z^{(0)}$ and $z^{(t-1)}$ ensures that each step is guided by both accumulated reasoning and the fixed problem context. After $T$ steps, the final latent state is decoded into a discrete prediction $y = f_{\text{tail}}(z^{(T)})$.

We define the sequence $\{z^{(1)}, z^{(2)}, \ldots, z^{(T)}\}$ as the model's cognitive trajectory. Each $z^{(t)}$ functions as an intermediate subgoal—an internal representation that is not merely a step toward the answer, but a meaningful reasoning state that can (and should) be explicitly optimized. This reflects the core principle of Flow CoT: *recursive reasoning should not be viewed as black-box repetition, but as a structured cognitive evolution*—where each step contributes a distinct phase in a progressive latent reasoning trajectory. Unfortunately, existing recursive methods often apply identical supervision (e.g., same output labels) to all iterations, ignoring the fact that earlier reasoning stages have limited representational capacity and distinct functional roles. This misalignment can distort latent states and harm downstream refinement—an issue we address in Section 3.3 through progressive distillation.

### 3.2 SCOUT: Stepwise Cognitive Optimization Using Teachers

While Flow CoT provides a theoretical formulation of latent, multi-iteration reasoning, current recursive methods [5, 7] lack training strategies that explicitly pursue this trajectory. In particular, most existing recursive approaches supervise all iterations using identical targets, ignoring the evolving representational capacity of each step—thus deviating from the core principle of Flow CoT.

To bridge this gap, we propose **SCOUT** (*Stepwise Cognitive Optimization Using Teachers*), a lightweight fine-tuning framework that equips pretrained LLMs with Flow CoT capabilities through recursive refinement. Building directly on the Flow CoT formulation, SCOUT adopts the same modular decomposition: a head block $f_{\text{head}}$, a recursive reasoning block $f_\theta$, and a tail block $f_{\text{tail}}$. As shown in Figure 2, the input is encoded once, the latent state is iteratively updated over $T$ steps via $f_\theta$, and the final output is decoded from the last state using $f_{\text{tail}}$.

While this architecture aligns with the Flow CoT formulation, effectively training the model to follow such a recursive trajectory introduces unique challenges. First, each reasoning step should be supervised in a way that reflects its current level of abstraction, rather than applying uniform supervision across all iterations. Second, the model must be able to incorporate reasoning from earlier steps without disrupting the pretrained architecture's data flow and attention structure. To address these issues, SCOUT introduces two key mechanisms—each integrated with minimal architectural modification:

- **Progressive Distillation:** As shown in Figure 2(a), each latent state $z^{(t)}$ is decoded and aligned with a teacher of matching capacity. Later steps are guided by stronger teachers (e.g., larger LLMs), enabling step-wise cognitive refinement while avoiding over-regularization in early stages. This is implemented via per-step KL divergence losses (see Section 3.3).

- **Retrospective Reasoning:** As shown in Figure 2(b), we introduce a cross-attention module that fuses the prior state $z^{(t-1)}$ into the current update. The model performs self-attention over the original latent state $z^{(0)}$, while attending to previous reasoning traces via cross-attention. This allows each step to selectively reuse earlier thoughts while maintaining input grounding (see Section 3.4).

Together, these mechanisms enable SCOUT to realize Flow CoT using only recursive application and fine-tuning—without requiring annotated intermediate steps or architecture-level modifications. SCOUT supports step-wise alignment, latent coherence, and efficient deployment within standard LLM pipelines.

### 3.3 Progressive Distillation

Flow CoT assumes that different reasoning steps reflect different levels of abstraction and cognitive complexity. Early steps may produce tentative or exploratory reasoning, while later steps are expected to perform more precise inference. To support this progression, we propose **progressive distillation**, a supervision strategy that aligns each step with a teacher model of matching capacity.

Unlike prior methods that apply the same supervision target at every iteration, we assign each latent state $z^{(t)}$ a soft target distribution $q^{(t)}$ generated by a step-specific teacher model. Formally, given an input $x$, we define:

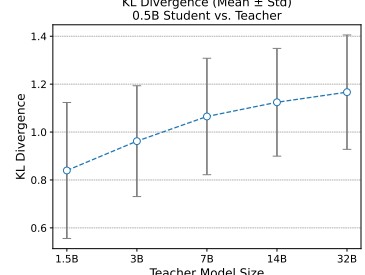

$$q^{(t)} = \text{Softmax}(f_{\text{teacher}_t}(x)), \tag{3}$$

where $f_{\text{teacher}_t}$ is a pretrained teacher LLM selected to match the expected capacity of step $t$. Early steps use smaller or distilled models (e.g., 0.5B, 1.5B), while later steps are supervised by increasingly powerful teachers (e.g., 7B, 14B, 32B), as illustrated in Figure 2(a).

Figure 3: KL divergence on Dolly dataset.

To empirically justify this approach, we measured the KL divergence between a student model (Qwen2.5-0.5B [41]) and a range of teacher models of increasing size, using 100 instructions sampled from the Dolly dataset [42]. As shown in Figure 3, the KL divergence increases steadily with teacher size. This indicates that larger teachers produce sharper and more complex distributions—containing richer learning signals. It also validates our hypothesis that step-specific supervision should be capacity-aware: stronger teachers should guide later steps that have greater representational strength.

For each reasoning step $t \in \{1, \dots, T\}$, the model produces a student distribution:

$$p_\theta^{(t)} = \text{Softmax}(f_{\text{tail}}(z^{(t)})), \tag{4}$$

which is optimized to match $q^{(t)}$ through a distillation loss:

$$\mathcal{L}^{(t)} = \text{KL}\left(q^{(t)} \,\|\, p_\theta^{(t)}\right) + \alpha \cdot \mathcal{L}_{\text{hard}}^{(t)}, \tag{5}$$

where the second term is an optional hard-label loss defined as:

$$\mathcal{L}_{\text{hard}}^{(t)} = \text{CE}(p_\theta^{(t)}, y^*), \tag{6}$$

with $y^*$ denoting the ground-truth output tokens and CE denoting cross-entropy.

Compared to uniform supervision, this strategy allows each step to optimize toward a learning target that matches its stage of cognitive development. This not only prevents over-regularization in early stages, but also ensures that stronger guidance is applied where the model is ready to absorb it. When access is limited to a single large model, one can first self-distill a size-stratified set of students and assign them to recursion steps from weakest to strongest, thereby preserving capacity matching.

### 3.4   Retrospective Reasoning Module

Recursive reasoning requires the model to build upon its prior cognitive states while staying grounded in the original input. However, pretrained LLMs are typically optimized for single-pass processing, where self-attention is computed over a fixed input sequence. Simply feeding previous outputs back into the prompt or modifying layer behavior often disrupts this pretrained flow—leading to suboptimal performance or the need for retraining [4].

To address this, we design a lightweight **retrospective reasoning module** that enables information flow across steps while minimally altering the model's original computation logic. Our design principle emphasizes simplicity and compatibility: each reasoning step reprocesses the original input via self-attention, while selectively attending to the prior step's latent state via cross-attention. This allows the model to treat its previous state $z^{(t-1)}$ as external memory, while maintaining stable computation over the static input embedding $z^{(0)}$.

Formally, we implement the history integration function $\mathcal{H}$ as introduced in Eq. (2), where each $z^{(t)}$ is updated by fusing $z^{(0)}$ and $z^{(t-1)}$ via a two-stream attention mechanism. Note that this mechanism is applied only for $t \geq 2$, as the initial step $z^{(1)}$ is computed directly from $z^{(0)}$ without retrospective integration, as defined in Eq. (1). Specifically, the model uses self-attention over the original state $z^{(0)}$, and cross-attention over the previous step's output $z^{(t-1)}$, as shown in Figure 2(b). This setup provides two key benefits: (1) it preserves the model's original dataflow over $z^{(0)}$, avoiding destructive interference with pretraining behavior; and (2) it allows flexible, content-based retrieval from the prior step, making the reasoning process both coherent and adaptive.

Compared to alternative fusion methods (e.g., additive mixing or gated updates [7, 4]), cross-attention offers a modular, query-driven mechanism. It preserves the representational separation between the current task context and past cognitive traces. We refer to this design as **XAttn** in subsequent analysis, to distinguish it from other integration strategies. Unlike methods that modify internal layer behavior [4], our approach introduces only a shallow wrapper—requiring neither architectural rewiring nor re-initialization of pretrained weights. This design aims to enable coherent multi-step reasoning while remaining fully compatible with pretrained LLMs.

### 3.5   Training and Inference

**Training.** SCOUT is trained via standard fine-tuning, but with supervision applied across $T$ *internal reasoning steps* rather than only the final output. In our experiments, $T$ is treated as a fixed hyper-parameter (e.g., $T = 3$), though it can be extended to dynamic reasoning schedules in future work. At each step $t \in \{1, \ldots, T\}$, the model refines its latent state $z^{(t)}$ using the retrospective module and recursive block (see Section 3.4). Each $z^{(t)}$ is decoded and supervised using a capacity-matched teacher, as described in Section 3.3. The total training loss is the weighted sum of per-step losses:

$$\mathcal{L} = \sum_{t=1}^{T} \lambda_t \cdot \mathcal{L}^{(t)}. \tag{7}$$

This formulation enables the model to learn step-wise cognitive refinement through end-to-end backpropagation, without architectural changes or pretraining modifications.

**Inference.** At inference time, the model follows the recursive update described in Eq. (1)–(2). It initializes $z^{(0)}$ via $f_{\text{head}}$, and iteratively updates the latent state across $T$ steps. The final state $z^{(T)}$ is decoded by $f_{\text{tail}}$ to produce the output. Optional early stopping can be applied based on output entropy or inter-step consistency.

**Efficiency.** SCOUT reuses existing model layers, adds only shallow cross-attention modules, and requires $T$ recursive passes per input. By preserving the model's original reasoning flow, it remains fully compatible with off-the-shelf LLMs and standard fine-tuning workflows.

## 4 Experiments

### 4.1 Experimental Setup

**Model architecture.** We use the Qwen2.5 series [43] and select Qwen2.5-0.5B as the student model. Recursive fine-tuning is applied under the SCOUT framework with $T = 3$ reasoning iterations. For progressive distillation, we assign Qwen2.5-1.5B, Qwen2.5-3B, and Qwen2.5-7B as teacher models for reasoning iterations 1, 2, and 3, respectively. See Appendix A.3 for additional implementation details.

**Training configuration.** We fine-tune the model for 2 epochs with a learning rate of $2 \times 10^{-5}$ using the LlamaFactory framework [44]. We fine-tune the model using a composite instruction-following dataset, which is a mix of five diverse instruction-tuning datasets. These datasets aim to improve the model's ability to follow specific instructions and reason through tasks step-by-step. The mixed dataset includes data from: (1) Alpaca GPT4 [45] and Alpaca CoT [46] (general instruction-following and chain-of-thought reasoning), (2) WikiQA [47] (open-domain question answering), (3) CodeAlpaca [48] (code generation), and (4) MathInstruct [14] (multi-step mathematical reasoning). For distillation, we apply the Adaptive Kullback-Leibler (AKL) [49] method to align student outputs with teacher distributions. Unless otherwise specified, we set $\lambda_t = 1/3$ for all $t$, assigning equal weight to each reasoning iteration. Further implementation details are provided in the Appendix A.1.

**Evaluation benchmarks.** To assess the performance of the fine-tuned model, we evaluate it using the lm-evaluation-harness framework [50] across a broad set of benchmarks, categorized into four areas: (i) *commonsense QA*, including ARC-easy, ARC-challenge [51], OpenBookQA [52], and TruthfulQA [53]; (ii) *multi-step reasoning*, such as GSM8K [54] and MMLU [55]; (iii) *reading comprehension and dialogue*, with CoQA [56] and GLUE [57]; and (iv) *code generation*, using MBPP [58]. These evaluation benchmarks are used to measure how well the model performs after fine-tuning, reflecting its reasoning ability across a range of domains.

**Baselines.** We compare SCOUT against a series of baselines to isolate the effect of recursive refinement and supervision strategies. All recursive variants share the same architecture based on simple layer stacking, ensuring that differences arise solely from how supervision is applied across iterations.

- **SFT (Supervised Fine-Tuning)**: Standard supervised fine-tuning on the same training data, where the loss is applied only at the final output, with no recursive refinement.
  **DSFT (Distilled-SFT)** [49]: Fine-tunes the model using soft targets from Qwen2.5-7B, but without any recursive computation.
- **R-SFT (Recursive Hard-Label Supervision)** [7]: Fine-tuning within the recursive framework, but with hard-label supervision at each step (i.e., each iteration applies standard supervised learning using ground-truth labels, without progressive distillation).
- **R-Distill-EQ (Recursive Distillation with Equal Weights)**: Uses the recursive architecture with a fixed 7B teacher for all iterations and applies equal loss weights $\lambda_1 = \lambda_2 = \lambda_3 = 1/3$ to provide uniform supervision.
- **R-Distill-WT (Recursive Distillation with Weighted Loss)** [8]: Similar to **R-Distill-EQ**, but applies increasing weights at each iteration to reflect the growing abstraction: $\lambda_1 = 0.2$, $\lambda_2 = 0.3$, $\lambda_3 = 0.5$, with stronger supervision on later reasoning stages.
- **R-SCOUT (Reversed SCOUT)**: A control variant where the teacher order is reversed (7B $\rightarrow$ 3B $\rightarrow$ 1.5B), violating the progressive distillation strategy and providing insights into the impact of teacher progression order.

### 4.2 Comparative Analysis with Baselines

We analyze the results in Table 1 by comparing SCOUT with a range of baseline methods, in order to assess the impact of recursive design and supervision strategies.

Table 1: Experimental results for different methods and iterations. $\Delta$ denotes the improvement relative to SFT; positive values are highlighted in blue, negative in red. *Iter.* denotes the number of recursive steps. All results are averaged across tasks. Both Avg and $\Delta$ values are computed before rounding, and then rounded to two decimal places. *Abbreviations: OB = OpenBookQA; ARC-e = ARC-Easy; ARC-c = ARC-Challenge; TF = TruthfulQA.*

| Method | Iter. | OB | GSM8K | MBPP | ARC-e | ARC-c | TF | CoQA | GLUE | Avg | $\Delta$ |
|---|---|---|---|---|---|---|---|---|---|---|---|
| SFT | - | 23.8 | 32.07 | 27.6 | 66.58 | 30.72 | 26.44 | 45.95 | 44.54 | 37.21 | - |
| DSFT | - | 23.8 | 33.59 | 27.4 | 64.94 | 27.47 | 26.81 | 48.97 | 51.55 | 38.06 | +0.85 |
| R-SFT | 1 | 24.6 | 31.84 | 27.4 | 66.08 | 31.83 | 25.83 | 44.62 | 47.72 | 37.49 | +0.27 |
|  | 2 | 25.2 | 32.22 | 27.0 | 64.94 | 32.08 | 26.93 | 44.88 | 48.61 | 37.73 | +0.52 |
|  | 3 | 24.6 | 32.22 | 27.0 | 64.31 | 32.34 | 26.81 | 45.40 | 47.28 | 37.49 | +0.28 |
| R-Distill-EQ | 1 | 23.8 | 35.10 | 28.0 | 65.95 | 28.24 | 26.68 | 47.80 | 52.02 | 38.44 | +1.23 |
|  | 2 | 23.8 | 33.81 | 26.2 | 63.64 | 28.50 | 26.81 | 46.77 | 47.84 | 37.17 | -0.04 |
|  | 3 | 22.2 | 33.59 | 27.4 | 63.30 | 28.16 | 26.93 | 47.85 | 49.49 | 37.36 | +0.15 |
| R-Distill-WT | 1 | 24.0 | 33.59 | 27.6 | 65.87 | 28.50 | 26.81 | 48.70 | 50.77 | 38.23 | +1.01 |
|  | 2 | 23.2 | 32.68 | 27.6 | 63.89 | 27.82 | 26.68 | 48.18 | 51.97 | 37.75 | +0.54 |
|  | 3 | 22.8 | 33.97 | 27.6 | 64.39 | 28.67 | 26.93 | 48.48 | 51.87 | 38.08 | +0.87 |
| R-SCOUT | 1 | 24.8 | 34.87 | 28.4 | 66.71 | 29.44 | 25.95 | 48.60 | 50.75 | 38.69 | +1.47 |
|  | 2 | 21.4 | 33.66 | 28.0 | 65.07 | 28.33 | 27.05 | 48.60 | 50.84 | 37.86 | +0.65 |
|  | 3 | 20.4 | 32.60 | 26.8 | 62.50 | 26.96 | 26.81 | 49.28 | 48.95 | 36.78 | -0.42 |
| SCOUT | 1 | 21.8 | 33.46 | 27.6 | 63.72 | 27.05 | 26.07 | 49.47 | 50.37 | 37.44 | +0.23 |
|  | 2 | 25.2 | 34.39 | 27.2 | 63.83 | 29.44 | 27.05 | 48.18 | 50.81 | 38.26 | +1.05 |
|  | 3 | 24.0 | 35.45 | 28.2 | 64.73 | 30.20 | 28.56 | 48.75 | 52.35 | **39.03** | **+1.81** |

**Distillation and recursion individually help, but remain limited.** DSFT, which applies soft targets from a strong teacher without recursion, achieves a modest improvement over SFT (+0.85), confirming that teacher signals improve calibration. Similarly, R-SFT and R-Distill-EQ add recursive structure by supervising each iteration independently. While these variants show initial gains (e.g., R-SFT: +0.27 → +0.52 → +0.28), performance stagnates or regresses in later steps (R-Distill-EQ: +1.23 → –0.04 → +0.15). This suggests that without step-specific guidance, recursion alone fails to support progressive refinement and can even introduce noise that hinders later-stage reasoning.

**Simple weighting helps—but only partially.** To mitigate early over-regularization, R-Distill-WT applies loss weights that increase across iterations ($\lambda_3 = 0.5$). This stabilizes later performance compared to R-Distill-EQ (e.g., +0.87 vs. +0.15 at step 3), but overall gains remain modest. Since all steps still rely on a fixed teacher (7B), the model cannot align supervision with its evolving capacity—highlighting that weighting alone cannot substitute for progressive teacher guidance.

**SCOUT achieves coherent, step-wise refinement.** SCOUT outperforms all baselines, with monotonic improvement across iterations (+0.23 → +1.05 → +1.81), and highest final performance (e.g., 35.5 on GSM8K, 52.4 on GLUE). These gains validate the Flow CoT principle: effective multi-step reasoning requires iteration-specific supervision and cognitive reuse. In contrast, R-SCOUT, which reverses teacher order (7B → 3B → 1.5B), initially achieves strong results (+1.47) but collapses in later steps (–0.42), confirming the necessity of aligning teacher strength with iteration depth.

Our findings reveal three insights: (1) recursion and distillation must be combined to support depth-aware reasoning; (2) fixed or reversed teacher strategies can degrade performance; and (3) SCOUT uniquely enables progressive cognitive refinement through modular architecture, aligned supervision, and cross-step integration.

### 4.3 Effect of Retrospective Integration Mechanisms

To evaluate the impact of retrospective reasoning design, we compare our cross-attention approach (**XAttn**) with five common alternatives: initial injection (**Init**) [5], additive fusion (**Add**) [4], concat-and-project (**CatProj**), gate-based control (**Gate**), and AdaLN-style modulated injection (**ModInj**). See Appendix B.3 for detailed implementations.

Table 2: Average accuracy across benchmarks using different retrospective modules.

| Iter. | Init | Add | CatProj | Gate | ModInj | XAttn (ours) |
|---|---|---|---|---|---|---|
| 1 | 37.18 | **37.59** | 36.36 | 33.85 | 34.55 | 37.49 |
| 2 | 34.10 | 35.38 | 27.75 | 27.94 | 32.16 | **37.73** |
| 3 | 27.11 | 29.09 | 23.95 | 27.09 | 28.98 | **37.50** |

Naive fusion strategies such as **Init**, **Add**, and **CatProj** show diminishing returns with deeper iterations, suggesting limited support for long-horizon reasoning. **ModInj** and **Gate** partially improve early performance but lack stability. Only our proposed **XAttn** yields consistent and monotonic improvements across all steps. This confirms the importance of query-driven, modular integration: **XAttn** enables cognitive coherence across iterations while preserving compatibility with pretrained inference flows—core to SCOUT's recursive design.

## 4.4 Qualitative Analysis: Iterative Cognitive Refinement

To better understand how SCOUT progressively refines its reasoning through recursive steps, we present two qualitative visualizations. These illustrate how SCOUT incrementally improves both its *answer accuracy* and *reasoning structure* through iteration.

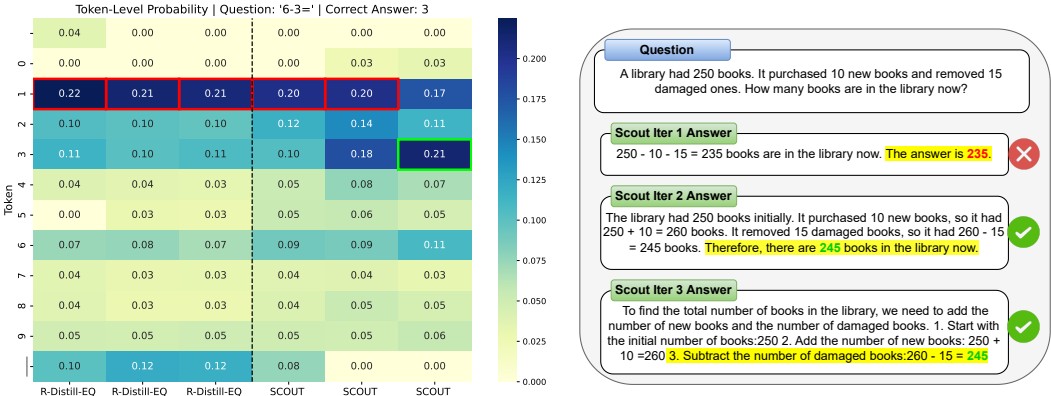

Figure 4: Token-level prediction probabilities on the math prompt "6 - 3 =". SCOUT gradually shifts probability mass toward the correct answer (**3**) across iterations, unlike the flat pattern in R-Distill-EQ.

Figure 5: Reasoning trace across SCOUT iterations for a book-counting question. The model transitions from incorrect shallow reasoning to accurate multi-step explanations.

**Refining belief through iteration.** Figure 4 presents a token-level probability heatmap for the math problem "6 - 3 =". The correct answer is 3. In R-Distill-EQ, all iterations assign high probability to token 1, and predictions stagnate. In contrast, SCOUT exhibits a distinct shift in attention: the first iteration mirrors R-Distill-EQ's output (1 dominates), but the second iteration begins to assign more weight to 3, and by the third iteration, 3 emerges as the top prediction. This trajectory illustrates how SCOUT performs latent search over cognitive space, adjusting its belief toward a more plausible output step-by-step.

**Improving reasoning quality across iterations.** Figure 5 showcases the evolution of SCOUT's natural language reasoning across three iterations on a simple arithmetic question. The first response is syntactically valid but mathematically incorrect (computes 250 - 10 - 15 = 235). The second iteration corrects the logic and provides a more faithful solution (260 - 15 = 245). The third response further improves by structuring the explanation into ordered sub-steps, reflecting improved decomposition and explanation clarity. This illustrates how recursive refinement in SCOUT not only improves accuracy, but also enriches the reasoning granularity and explanatory quality.

These examples confirm that SCOUT's iterative mechanism is not merely repeating inference but actively refining both the *belief space* (token predictions) and the *reasoning space* (explanation quality) over time—embodying the core principle of Flow CoT.

# 5  Conclusion

We present **Flow Chain-of-Thought (Flow CoT)**, a reasoning paradigm that models multi-step inference as a progressive trajectory of latent cognitive states. Unlike traditional Chain-of-Thought prompting, which relies on manually curated step-by-step traces, Flow CoT enables reasoning to emerge organically through recursive computation—offering a scalable and cognitively grounded alternative. To instantiate this paradigm, we introduce **SCOUT**, a lightweight fine-tuning framework that equips pretrained LLMs with Flow CoT capabilities. SCOUT combines progressive distillation—where each reasoning step is supervised by a capacity-matched teacher—with a cross-attention-based retrospective module that enables selective integration of prior outputs. Importantly, SCOUT requires no additional pretraining. Across eight benchmarks, SCOUT consistently improves both final answer accuracy and explanation quality, validating the effectiveness and practicality of Flow CoT under a fine-tuning regime.

**Limitations and Future Work.** SCOUT currently uses a fixed number of reasoning steps and manually selected teacher models, which provide clarity and training stability but may limit adaptability to tasks of varying complexity. Future work could explore dynamic iteration control or adaptive teacher selection, for example through reinforcement-learning–based scoring. More broadly, as Flow CoT is intended as a general reasoning framework, it opens opportunities for new architectures and training strategies that support structured, cognitively motivated inference. We hope this work encourages further exploration of principled recursion in language model design.

# Acknowledgments

This work was supported by the National Key R&D Program of China (2022YFB4701400/4701402), SSTIC Grants (KJZD20230923115106012, KJZD20230923114916032, GJHZ20240218113604008), the National Natural Science Foundation of China under Grant (62202214), and the Guangdong Basic and Applied Basic Research Foundation under Grant (2023A1515012819).

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

# Appendix

This appendix provides full implementation and evaluation details to complement the main paper. We include training configurations, dataset statistics, benchmark setups, model partitioning strategies, and ablation results for various retrospective integration mechanisms.

## A  Additional Experimental Details

### A.1  Training Settings

**Hardware.**  All experiments are conducted on a single **NVIDIA H20 NVLink** GPU (96 GB) attached to a dual–socket server with **20 CPU cores** (Intel® Xeon® Platinum 8457C) and **200 GB** RAM. We utilize `torch` native *gradient accumulation* to emulate a global batch size of 128 sequences.

**Optimization hyper-parameters.**  Below we detail the learning-rate schedule and other relevant knobs. Fine-tuning lasts for 2 epochs with a learning rate of $\mathrm{lr_{pre}} = 2 \times 10^{-5}$ applied to all *pretrained* parameters. Newly introduced parameters (e.g., cross-attention projection, FC adapters, layer-norm gates, etc.) are trained with a higher learning rate $\mathrm{lr_{new}} = 2\mathrm{lr_{pre}}$ to accelerate adaptation while minimizing residual drift. We employ a **cosine learning rate schedule** with a **warm-up ratio of 10%**. Training is performed using **bf16 precision**.

**Distillation loss.**  Unless stated otherwise, the per-iteration objective is defined as

$$\mathcal{L}^{(t)} = \mathrm{KL}\big(q^{(t)} \parallel p_\theta^{(t)}\big) + \alpha\, \mathcal{L}_{\mathrm{hard}}^{(t)}, \tag{8}$$

where we use $\alpha = 0.5$. This balances soft-target alignment with ground-truth cross-entropy, matching the setting that yields the best validation perplexity in a $3 \times 3$ grid search over $\alpha \in \{0.3, 0.5, 0.7\}$.

### A.2  Fine-tuning Data

Table 3 summarizes the five instruction-tuning corpora used for SCOUT. We follow the official splits and concatenate them after normalizing to a conversational format compatible with `Qwen2.5`.

Table 3: Statistics of the composite instruction corpus.

| Dataset | Domain | Purpose |
|---|---|---|
| Alpaca GPT-4 [45] | general | single-turn instructions; broad world knowledge |
| Alpaca CoT [46] | reasoning | curated chain-of-thought exemplars to expose stepwise patterns |
| WikiQA [47] | open-domain QA | factual span extraction, grounding in Wikipedia context |
| CodeAlpaca [48] | code | code synthesis / completion to exercise symbolic reasoning |
| MathInstruct [14] | maths | multi-step numerical reasoning, explicit intermediate variables |

### A.3  Model Design

**Choice of Backbone and Teachers.**  We adopt the Qwen2.5 family because it provides a size-stratified yet homogeneous suite in both architecture and training recipe (0.5B, 1.5B, 3B, 7B, . . . ), enabling capacity-matched progressive distillation while minimizing confounders across teacher capacities. Qwen2.5 also enjoys broad community adoption and strong reproducibility.

**Partition strategy.**  Unless otherwise specified, we adopt the following configuration—referred to as **Case 2** in Table 5:

- `Embedding + first 1/2 layers` → Head block
- `Remaining 1/2 layers` → Recursive block
- `Output projection` → Tail block

This layout preserves deeper semantics in the head and allocates more capacity to recursive reasoning, as later confirmed in our ablation results.

For ablation analysis (Appendix B.2), we compare this against the following alternative configuration—referred to as **Case 1**:

- `Embedding + first 1/3 layers` → `Head`
- `Middle 1/3 layers` → `Recursive`
- `Final 1/3 layers + FC` → `Tail`

**Cross-attention details.** Each recursive block layer is augmented with a *causal* cross-attention sub-layer that queries the previous iteration's output hidden states. We reuse the same causal mask as self-attention, ensuring no access to future tokens. We also insert a two-layer MLP projection before the cross-attention module to map the current representation into the key/value space of the previous step, which helps stabilize training and improves convergence.

**Vocabulary compatibility.** The teacher Qwen2.5-7B model uses $|V| = 152{,}064$ sub-word tokens, whereas the student 0.5B model has $|V| = 151{,}936$. The extra 128 symbols are all `<special_added_xx>` placeholders. During distillation, we **(i)** truncate the teacher logits to the student vocabulary size and **(ii)** renormalize before the softmax, so that the probability mass on absent IDs is redistributed proportionally.

## A.4 Retrospective Reasoning Module

To support information flow across reasoning steps without disrupting the structure of pretrained LLMs, we introduce a lightweight *retrospective module* that fuses each latent state $z^{(t-1)}$ with the initial state $z^{(0)}$ through a history integration function $\mathcal{H}(z^{(0)}, z^{(t-1)})$, producing the contextual input for the current step.

We compare six retrospective integration mechanisms—five from prior work and one proposed by us—under the standard R-SFT setting (i.e., hard-label supervision at each recursion step). These include:

- **Init injection (Init)** [7], Reuse the previous latent state directly: $\mathcal{H}(z^{(0)}, z^{(t-1)}) = z^{(t-1)}$.
- **Additive fusion (Add)** [4], Combine prior and initial states by addition: $\mathcal{H}(z^{(0)}, z^{(t-1)}) = z^{(0)} + z^{(t-1)}$.
- **Gate-based control (Gate)** [4], Use a learnable gate to modulate the contribution of $z^{(t-1)}$ via element-wise control.
- **Concat-and-project (CatProj)** [4], Concatenate the states and apply a projection: $\mathcal{H}(z^{(0)}, z^{(t-1)}) = \text{MLP}([z^{(0)}; z^{(t-1)}])$.
- **Modulated injection (ModInj)** [4], Inject $z^{(t-1)}$ into the current computation via AdaLN-style modulation.
- **Cross-attention (XAttn, ours)**, Treat $z^{(t-1)}$ as external memory and apply cross-attention using $z^{(0)}$ as the query. This allows selective retrieval of prior reasoning while preserving the model's original attention flow.

## A.5 Evaluation Benchmarks

We evaluate model performance using `lm-evaluation-harness` [50], with metrics summarised below for each dataset.

- **Commonsense QA.** *ARC-easy/challenge* [51] and *OpenBookQA* [52] are multiple-choice datasets; their accuracy are reported. *TruthfulQA* [53] is evaluated with the MC1-single accuracy variant.
- **Multi-step reasoning.** *GSM8K* [54] measures mathematically-grounded free-form answers via exact string match.

- **Reading comprehension & dialogue.** *CoQA*[56] is assessed with F1 over answer spans. *GLUE* [57] is the nine-task natural-language-understanding suite; we follow the harness convention and report the unweighted average of dev-set metrics (e.g., MNLI-m accuracy, QQP F1).
- **Code generation.** *MBPP* [58] is evaluated with the pass-@-1 criterion using harness-built execution-based tests.

# B   More Experimental Results

This appendix extends Section 4 with additional analyses, including: progressive distillation (single-pass, no recursion), ablations, dataset-level breakdowns, and module comparisons to further validate our approach. Unless otherwise noted, averages are computed before rounding and then rounded to two decimals.

## B.1   Progressive Distillation (Single-Pass, No Recursion)

Table 4: One-pass progressive distillation of a 0.5B student from 1.5B→3B→7B teachers).

| Model | OB | GSM8K | MBPP | ARC-e | ARC-c | TF | CoQA | GLUE | Avg |
|---|---|---|---|---|---|---|---|---|---|
| Progressive | 24.4 | 32.37 | 27.2 | 64.73 | 27.39 | 29.01 | 48.28 | 51.08 | 38.06 |

We distill a 0.5B student sequentially from 1.5B $\to$ 3B $\to$ 7B teachers in a single pass (no recursive reasoning). This simple baseline attains performance comparable to direct distillation from the 7B teacher. Without iterative reasoning, the student tends to retain primarily the last (largest) teacher's signal, whereas SCOUT improves with each recursion step via stepwise supervision and history-aware refinement.

## B.2   Ablation on structural partitioning

Table 5: **Ablation on structural partitioning. Case 1** = *Embedding + first $\frac{1}{3}$ layers as head, middle $\frac{1}{3}$ layers as recursive block, final $\frac{1}{3}$ layers + output-projection (FC) as tail*; **Case 2** = *Embedding + first $\frac{1}{2}$ layers as head, remaining $\frac{1}{2}$ layers as recursive block, output-projection as tail*. For each iteration, $\Delta$ reports the gain of Case 2 over the corresponding Case 1 (blue = positive). All results are obtained under the R-SFT regime, using hard-label supervision at each recursive step. OB = OpenBookQA; ARC-e = ARC-Easy; ARC-c = ARC-Challenge; TF = TruthfulQA.

| Method | Iter. | OB | GSM8K | MBPP | ARC-e | ARC-c | TF | CoQA | GLUE | Avg | $\Delta$ |
|---|---|---|---|---|---|---|---|---|---|---|---|
| **Init inj. – Case 1** | 1 | 22.40 | 16.38 | 15.80 | 61.62 | 28.41 | 25.21 | 40.33 | 41.25 | 31.43 | – |
|  | 2 | 21.20 | 16.07 | 7.40 | 59.93 | 29.27 | 24.24 | 35.73 | 44.06 | 29.74 | – |
|  | 3 | 21.00 | 8.49 | 0.60 | 56.78 | 28.58 | 24.24 | 24.92 | 46.97 | 26.45 | – |
| **Init inj. – Case 2** | 1 | 24.80 | 30.55 | 26.20 | 65.74 | 31.23 | 25.83 | 48.20 | 44.86 | 37.18 | +5.75 |
|  | 2 | 23.40 | 26.31 | 26.20 | 64.18 | 31.23 | 27.78 | 30.40 | 43.33 | 34.10 | +4.37 |
|  | 3 | 25.20 | 14.86 | 13.80 | 59.09 | 31.06 | 28.15 | 2.15 | 42.55 | 27.11 | +0.66 |
| **Add. fusion – Case 1** | 1 | 21.40 | 1.97 | 0.20 | 57.32 | 27.13 | 22.15 | 8.35 | 43.88 | 22.80 | – |
|  | 2 | 19.20 | 1.97 | 0.00 | 56.36 | 26.54 | 23.50 | 9.93 | 44.55 | 22.76 | – |
|  | 3 | 18.00 | 1.67 | 0.00 | 54.29 | 25.43 | 24.24 | 11.75 | 45.94 | 22.67 | – |
| **Add. fusion – Case 2** | 1 | 25.20 | 31.16 | 28.20 | 66.54 | 30.29 | 24.60 | 47.52 | 47.23 | 37.59 | +14.79 |
|  | 2 | 24.60 | 29.57 | 24.80 | 64.23 | 32.42 | 26.93 | 35.97 | 44.53 | 35.38 | +12.63 |
|  | 3 | 23.60 | 17.36 | 13.80 | 60.06 | 33.36 | 27.78 | 14.22 | 42.56 | 29.09 | +6.43 |
| **Cross-attn – Case 1** | 1 | 20.20 | 1.36 | 0.00 | 55.09 | 27.56 | 23.99 | 7.33 | 48.58 | 23.01 | – |
|  | 2 | 18.40 | 1.44 | 0.00 | 50.88 | 24.91 | 24.60 | 4.77 | 41.88 | 20.86 | – |
|  | 3 | 18.20 | 1.29 | 0.00 | 51.64 | 24.83 | 24.48 | 5.93 | 45.79 | 21.52 | – |
| **Cross-attn – Case 2** | 1 | 24.60 | 31.84 | 27.40 | 66.08 | 31.83 | 25.83 | 44.62 | 47.72 | 37.49 | +14.48 |
|  | 2 | 25.20 | 32.22 | 27.00 | 64.94 | 32.08 | 26.93 | 44.88 | 48.61 | 37.73 | +16.87 |
|  | 3 | 24.60 | 32.22 | 27.00 | 64.31 | 32.34 | 26.81 | 45.40 | 47.28 | 37.50 | +15.98 |

**Analysis of Layer-Partition Ablations.** To systematically examine the impact of structural partitioning, we compare two layout strategies (Case 1 vs. Case 2) under three different retrospective modules—Init injection, Additive fusion, and Cross-attention (ours)—within the R-SFT training regime. That is, each recursive step uses standard supervised learning with ground-truth labels, without progressive distillation. The goal is to isolate the effect of partition strategy while holding training and integration conditions fixed.

Table 5 presents the results. We interpret them based on two empirical facts: (i) pretrained layer representations are *not* uniformly informative—early layers encode low-level lexical cues, middle layers gradually disentangle semantics, and late layers specialize in reasoning and decoding; (ii) recursive refinement can only add information that the recursive block itself is capable of modeling.

- **Larger heads expose richer priors.** Case 2 promotes half of the backbone to the head block, giving the very first forward pass direct access to deeper semantic features that are otherwise reached only after recursion in Case 1. Consequently, iteration 1 already outperforms the entire Case-1 trajectory by +14.5–14.8 points.

- **Recursive depth has diminishing returns when the middle block is shallow.** In Case 1 the recursive block consists of merely $\frac{1}{3}$ of the layers; its latent states ($z^{(t)}$) carry limited new evidence, so iteration 3 accrues noise faster than signal. Expanding the recursive block to $\frac{1}{2}$ (Case 2) stabilizes iteration 2, but iteration 3 still plateaus—suggesting that once high-capacity semantics are already present in $z^{(1)}$, further passes yield marginal benefit.

- **Cross-attention maximises structural capacity.** The XAttn integration module exploits the richer head features without disrupting the pretrained attention flow. By treating the previous latent state as an external memory, it filters useful cues while discarding spurious ones, giving the largest average gain (+16.9 at $t=2$) under the same parameter budget.

- **Why "half–half–0" offers stronger performance.** Pretraining concentrates reasoning ability toward later layers; shifting those layers to the head (**1/2**) exposes them to the recursive block early. The recursive block—now also **1/2**—receives stronger inputs and therefore produces more meaningful refinements, which the lightweight tail (FC) can decode with minimal overhead. In essence, Case 2 turns recursion into *iterative polishing of already high-quality embeddings*, whereas Case 1 must first build that quality through shallow recursion, incurring greater noise amplification.

- **Limitations and future work.** Owing to computational constraints, we did not explore finer-grained or alternative layer-allocation strategies. Systematically varying the head/recursive/tail split—especially with dynamic or data-adaptive partitions—remains an interesting direction for future research.

Overall, these findings substantiate our design choice: assigning more pretrained depth to the head and recursive blocks, coupled with cross-step attention, yields the most faithful instantiation of *Flow CoT*—achieving faster convergence and higher final accuracy without extra parameters or pretraining.

### B.3 Dataset-Level Breakdown of Retrospective Modules

This section complements the integration strategies described in Appendix A.4 by reporting dataset-level performance across recursion depths ($t = 1, 2, 3$). All results are obtained under the R-SFT training regime [7], where each iteration uses standard hard-label supervision without progressive distillation. Table 6 presents detailed benchmark scores for each retrospective module. All metrics are reported as **accuracy** (%), except for *CoQA* (F1) and *GLUE* (average).

**Per-dataset observations.**

- **OpenBookQA.** XAttn obtains the highest score (25.2) at $t=2$, narrowly surpassing Additive fusion (25.2 at $t=1$) while maintaining stability at deeper steps.

- **GSM8K (math).** XAttn leads on every iteration, peaking at 32.22. CatProj collapses after the first step, revealing its difficulty in propagating structured numeric information.

- **MBPP (code).** Additive fusion yields the single best result (28.2) but deteriorates sharply afterwards; XAttn retains $>27$ across all iterations, indicating more robust generalisation.

Table 6: Detailed benchmark results for different retrospective modules.

| Module | Iter. | OB | GSM8K | MBPP | ARC-e | ARC-c | TF | CoQA | GLUE | Avg |
|---|---|---|---|---|---|---|---|---|---|---|
| Init | 1 | 24.8 | 30.55 | 26.20 | 65.74 | 31.23 | 25.83 | 48.20 | 44.86 | 37.18 |
|  | 2 | 23.4 | 26.31 | 26.20 | 64.18 | 31.23 | 27.78 | 30.40 | 43.33 | 34.10 |
|  | 3 | 25.2 | 14.86 | 13.80 | 59.09 | 31.06 | 28.15 | 2.15 | 42.55 | 27.11 |
| Add | 1 | 25.2 | 31.16 | 28.20 | 66.54 | 30.29 | 24.60 | 47.52 | 47.23 | 37.59 |
|  | 2 | 24.6 | 29.57 | 24.80 | 64.23 | 32.42 | 26.93 | 35.97 | 44.53 | 35.38 |
|  | 3 | 23.6 | 17.36 | 13.80 | 60.06 | 33.36 | 27.78 | 14.22 | 42.56 | 29.09 |
| CatProj | 1 | 23.0 | 32.37 | 24.00 | 63.93 | 30.38 | 23.75 | 45.60 | 47.85 | 36.36 |
|  | 2 | 20.6 | 20.62 | 0.20 | 53.91 | 26.28 | 25.83 | 27.67 | 46.89 | 27.75 |
|  | 3 | 20.0 | 7.96 | 0.00 | 49.49 | 26.11 | 24.48 | 19.83 | 43.69 | 23.95 |
| Gate | 1 | 24.0 | 27.90 | 7.60 | 65.61 | 29.61 | 25.58 | 39.42 | 51.08 | 33.85 |
|  | 2 | 17.0 | 22.52 | 0.40 | 56.86 | 26.45 | 24.85 | 25.33 | 50.08 | 27.94 |
|  | 3 | 20.0 | 15.69 | 1.20 | 60.14 | 27.39 | 25.83 | 24.25 | 42.18 | 27.09 |
| ModInj | 1 | 21.2 | 28.81 | 23.00 | 63.01 | 28.58 | 25.46 | 40.53 | 45.79 | 34.55 |
|  | 2 | 20.8 | 24.03 | 16.20 | 61.91 | 29.61 | 25.09 | 36.78 | 42.89 | 32.16 |
|  | 3 | 20.6 | 16.07 | 8.60 | 59.09 | 28.16 | 25.34 | 29.43 | 44.57 | 28.98 |
| XAttn (ours) | 1 | 24.6 | 31.84 | 27.40 | 66.08 | 31.83 | 25.83 | 44.62 | 47.72 | 37.49 |
|  | 2 | 25.2 | 32.22 | 27.00 | 64.94 | 32.08 | 26.93 | 44.88 | 48.61 | 37.73 |
|  | 3 | 24.6 | 32.22 | 27.00 | 64.31 | 32.34 | 26.81 | 45.40 | 47.28 | 37.50 |

- **ARC-easy / ARC-challenge.** XAttn delivers the top or second-best accuracy on both ARC splits, confirming that cross-step attention is especially beneficial for commonsense and scientific QA.

- **TruthfulQA.** XAttn reaches 26.93, improving over Gate/ModInj by ∼1.4 pts, which suggests better resistance to hallucination when knowledge is revisited recursively.

- **CoQA.** Additive fusion attains the absolute maximum (47.52), yet performance drops by 33 pts at $t=3$. XAttn grows steadily to 45.40, demonstrating superior long-horizon dialogue coherence.

- **GLUE.** Gate-based control wins the first iteration (51.08) thanks to its explicit gating, but XAttn overtakes it at deeper steps (47.28 vs. 42.18) while keeping higher overall average.

**Take-aways.** Across eight diverse benchmarks, **XAttn is the only mechanism that (i) never collapses with depth and (ii) ranks first or second on *all* tasks**. Among the integration strategies introduced earlier, cross-attention clearly stands out. The results reinforce our main claim: *query-driven cross-attention provides the right inductive bias for reusing intermediate cognition*, whereas simpler fusions either over-mix (Add) or under-utilize (CatProj) the latent history. Future work could combine the strong first-step performance of Gate with XAttn's depth stability—for instance, by conditioning the cross-attention weights on a learnable gating signal.

## C   Impact Statement

Our work introduces *Flow Chain-of-Thought* (Flow CoT) and its practical instantiation *SCOUT*, enabling pretrained language models to achieve deeper, interpretable reasoning with *fine-tuning only*. Because SCOUT reuses existing weights and adds only lightweight cross-attention adapters, it lowers the computational and energy cost typically associated with multi-step reasoning systems that require extensive pre- or re-training. This can democratise access to stronger LLM reasoning in research, education, and low-resource settings while reducing the environmental footprint of model development.

As with any technique that amplifies model capability, Flow CoT may inadvertently strengthen harmful content generation, persuasive misinformation, or biased reasoning chains. Moreover, progressive distillation leverages larger teacher models whose biases may propagate to students. We encourage follow-up work to pair Flow CoT with robust safety filters, bias audits, and energy-usage tracking. In addition, dynamic early-exit policies could further reduce inference cost when high-confidence answers are reached, aligning computational efficiency with responsible AI principles.

