# OpenReview forum: "SCOUT: Teaching Pre-trained Language Models to Enhance Reasoning via Flow Chain-of-Thought"
_NeurIPS.cc/2025/Conference — NeurIPS 2025 poster_

### Official Review · Reviewer_Z32A · 2025-07-03

**Clarity:** 3
**Significance:** 3
**Originality:** 3
**Rating:** 5
**Confidence:** 3

**Summary:**

The paper proposes a flow CoT for recursive reasoning paradigm with depth-aware trajectory of latent cognitive states.  The authors develop a light-weight fine-tuning framework with a step wise optimization technique using teachers.  The introduction of a progressive distillation strategy enables the reasoning to be supervised by a teacher. The paper also presents a non-intrusive retrospective module using cross-attention that ensures the original reasoning flow. The authors empirically show the improvements in the performance on multiple benchmarks.

**Questions:**

1. The authors should clearly mention the reason for the model preference.

**Ethical Concerns:**

["NO or VERY MINOR ethics concerns only"]

**Final Justification:**

Yes the authors have addressed all the questions raised by me and I have given the scores accordingly.

**Limitations:**

Yes

**Quality:**

3

**Strengths And Weaknesses:**

The major strength of the paper lies in its capabilities of multi-step reasoning without requiring explicit annotations at each step. The paper attempts to train the model through progressive distillation using teachers with matching capacity unlike distillation requiring strong uniform supervision. The proposed retrospective module enhances depth-aware reasoning capabilities in pre-trained LLMs. The authors present a comparative study of the performance of the proposed model with different baselines. The token level probabilities heatmap shows a clear progressive trajectory towards the correct answer using the approach.

It is not clear why Qwen 2.5 model series is chosen for the task. Similarly if Qwen is used, the reason behind not using an update Qwen model is also missing.

---

> ### Author Rebuttal · Authors · 2025-07-31
>
> We thank you for the careful evaluation. Below we clarify our choice of the Qwen 2.5 series:
>
> 1. **Homogeneous Multi-Scale Model Family**
>    Qwen 2.5 provides a spectrum of models—from 0.5B, 1.5B, 3B, 7B up to 14B, 32B, and 72B—all trained with the same recipe. This architectural and training-data homogeneity allows us to perform progressive distillation on a single student model using increasingly stronger teachers, without introducing confounding variables.
>
> 2. **Community Adoption & Experimental Timeline**
>    Qwen 2.5 has been widely adopted in recent large-model research (e.g., [1], [2], [3]), demonstrating strong community support and reproducibility.  Although Qwen 3.0 was released at the end of April 2025, our experimental pipeline had already been finalized and the core experiments were nearly complete by then, making it infeasible to re-run the full study before the submission deadline.
>
>  We will include this rationale in the final manuscript. We hope this explanation clarifies our choice of the Qwen 2.5 series and appreciate your valuable feedback.
>
> ---
>
> >### References
> >1. He, Yichen *et al.* “PaSa: An LLM Agent for Comprehensive Academic Paper Search.” *Proceedings of the 63rd Annual Meeting of the Association for Computational Linguistics (Volume 1: Long Papers)*, 2025.
> >2. Hong, Jiwoo *et al.* “On the Robustness of Reward Models for Language Model Alignment.” *arXiv preprint arXiv:2505.07271*, 2025.
> >3. Yang, Shiming *et al.* “Demystifying Long Chain-of-Thought Reasoning.” *Proceedings of the Forty-second International Conference on Machine Learning*, 2025.

---

> > ### Comment · Reviewer_Z32A · 2025-08-05
> > **Response to Author Rebuttal**
> >
> > I have reviewed the author rebuttal and the authors have satisfactorily answered my query on using the Qwen 2.5 series models, not Qwen 3 models.

---

> > > ### Author Response · Authors · 2025-08-05
> > >
> > > Thank you for confirming that our explanation addressed your concern.   We will integrate the rationale for choosing the Qwen-2.5 series into the revised manuscript.  We appreciate your constructive feedback and the time you have devoted to our work.

---

### Official Review · Reviewer_rHmT · 2025-07-03

**Clarity:** 2
**Significance:** 3
**Originality:** 3
**Rating:** 4
**Confidence:** 3

**Summary:**

This paper points out a problem in existing recursive reasoning models: recursive reasoning should not be viewed as black-box repetition, but as a structured cognitive evolution. It introduces Flow CoT, a reasoning paradigm that envisions recursive inference in LLMs as a progressive trajectory of cognitive states, moving beyond conventional chain-of-thought, which depends on explicit supervision for intermediate steps. To operationalize this paradigm, the authors propose SCOUT, a fine-tuning framework that combines progressive distillation with a cross-attention-based retrospective reasoning module.  The approach is extensively evaluated on 8 benchmarks (covering commonsense QA, multi-step math, code, etc.), demonstrating improved accuracy and explanation quality versus several recent baselines.

**Questions:**

The concept of Flow CoT appears to share similarities with diffusion models. Is there a deeper connection between language diffusion models and your work? Could you elaborate on this relationship?

In Line 131, you state, "This misalignment can distort latent states and harm downstream refinement." This is a strong claim. Do you have any citations or concrete analyses to support this strong claim about existing recursive methods?

Some aspects of the method are still unclear. For instance, how are the head block and tail block explicitly defined when working with LLMs? Could you provide more details about the training and inference processes?

Could you explain why CoQA performance decreases as the number of reasoning steps increases?

Benchmarks such as GLUE, CoQA, and MBPP are generally not considered to require extensive reasoning. As a result, they may not be suitable for evaluating a model’s multi-step reasoning abilities. Can you explain more about the model behavior of these datasets and why they can be benefited from SCOUT?

**Ethical Concerns:**

["NO or VERY MINOR ethics concerns only"]

**Final Justification:**

The scaling experiments show positive results, and it is convincing to me.

**Limitations:**

Yes.

**Quality:**

2

**Strengths And Weaknesses:**

Strengths:
(1) Novel Concept: The paper introduces the concept of Progressive Distillation and leverages latent variables to enhance performance on reasoning tasks. This is a valuable contribution to the reasoning research community.

(2) Comprehensive Evaluation: The proposed SCOUT method is evaluated on eight benchmarks covering various types of tasks. The paper also explores different training settings and provides insights into the key aspects of the method.


Weaknesses:
(1) Scalability: The method depends on multiple teacher models with different abilities, raising concerns about scalability—especially when applying it to larger models. For very strong models, it may be difficult to find appropriately leveled teacher models for supervision. Additionally, experiments only go up to a 7B teacher model, which limits confidence in the method’s effectiveness at larger scales (e.g., 72B). The approach should be validated on a wider range of model sizes.

(2)Baseline Comparison: An additional baseline should be included where a single model is distilled progressively using teachers of sizes 1.5B to 7B in sequence, with inference performed only once (to distinguish from R-Distill-EQ). Also, the fair comparison for SCOUT should be with CoT-style SFT, not just pure SFT as currently reported. Comparisons with latent CoT-based methods would also strengthen the evaluation. At least the differnet should be discussed thoroughly.

(3) Lack of Efficiency Analysis: Although the method could offer efficiency advantages over traditional explicit reasoning approaches like R1, there is insufficient analysis of wall-clock time and inference speed. In particular, Flow CoT should theoretically be faster than decoding a model three times in a standard CoT setup, but this is not discussed.

---

> ### Author Rebuttal · Authors · 2025-07-31
>
> **Thank you for your detailed review. We address your concerns as follows:**
>
> ---
>
> ### **1. Scalability**
> #### **1.1 Large-Scale Evaluation**
> To address the suggestion for larger‐scale evaluation, we replaced the original 7B teacher with 14B and 32B model. The results are as follows:
>
> |Teacher Size|Iteration|OB|GSM8K|MBPP|ARC-e|ARC-c|TF|CoQA|GLUE|**Avg**|
> |:-:|:-:|:-:|:-:|:-:|:-:|:-:|:-:|:-:|:-:|:-:|
> |**14B** |1|22.6|33.21|27.2|65.49|28.67|26.68|49.63|49.27|37.84|
> ||2|22.4|34.27|28.4|65.49|29.18|26.81|47.33|52.68|38.32|
> ||3|24.0|34.39|28.4|66.54|30.29|27.42|47.80|54.41|39.16|
> |**32B**|1|23.4|33.74|27.8|66.62| 30.29 | 27.17| 48.32|52.22|38.70|
> ||2|23.2|34.27|28.0|66.88|30.63|27.17|47.05|53.58|38.85|
> ||3|24.0|34.72|29.0|66.96|30.72|27.66|47.12|54.05|39.28|
>
>  We observe that even with a 32B teacher, SCOUT’s average accuracy steadily improves over iterations (38.70→38.85→39.28), demonstrating both the effectiveness and scalability of our method at larger scales.
>
> #### **1.2 Teacher Models**
>
> Our central contribution is the Flow CoT paradigm—treating recursive reasoning as a progressive trajectory of latent states. SCOUT’s multi-teacher design is one concrete instantiation, not a prerequisite of Flow CoT, and our experimental results already demonstrate the effectiveness and scalability of both SCOUT and Flow CoT.
>
> We have also considered the point you raised.  Consequently, in the Limitations and Future Work section we explicitly state that the present study offers an  proof-of-concept validation of Flow CoT, and we outline alternative directions—such as *reinforcement-learning–based scoring without external teachers*—that can provide per-iteration difficulty rewards and represent promising avenues for future research.
>
>  ---
>
>  ### **2. Baseline Comparison**
>  Because SCOUT is designed for iterative models, our main experiments primarily compare against other recursive approaches to demonstrate the effectiveness of our method.  The additional baselines you suggested are conventional single-pass fine-tuning techniques that do not involve model recursion; nevertheless, we have included them for completeness.
>
> #### **2.1 Progressive Distillation Baseline**
> We added a progressive distillation baseline: a 0.5B student distilled sequentially from 1.5B→3B→7B teachers, yielding:
>
> |Model|OB|GSM8K|MBPP|ARC-e|ARC-c|TF|CoQA|GLUE|**Avg**|
> |-|:-:|:-:|:-:|:-:|:-:|:-:|:-:|:-:|:-:|
> |**Progressive**|24.4|32.37|27.2|64.73|27.39|29.01|48.28|51.08|38.06|
>
> We observe that simple, one-pass progressive distillation achieves a performance comparable to directly distilling with the 7B teacher. Without iterative reasoning, the student model tends to “forget” earlier-teacher knowledge and retain primarily the largest teacher’s knowledge—contrasting with SCOUT’s iterative improvement of reasoning capabilities.
>
> #### **2.2 Comparison with CoT-Style SFT**
> To compare against a chain-of-thought supervised fine-tuning (CoT-SFT) baseline, we evaluated both CoT-SFT and SCOUT on the AQUA-RAT math dataset:
>
> |Method|AQUA-RAT|
> |-|:-:|
> |**CoT-SFT**| 33.07    |
> |**SCOUT (Iter 1)**|32.28|
> |**SCOUT (Iter 2)**|33.46|
> |**SCOUT (Iter 3)**|**34.64**|
>
> SCOUT demonstrates consistent iterative gains, ultimately surpassing the CoT-SFT baseline by iteration three.
>
> #### **2.3 Comparison with Latent CoT Methods**
> - **Latent CoT (e.g., Coconut[1])** performs **horizontal recursion**: it feeds the last‐layer hidden state back into the model for the next step, requiring specialized data pipelines.
> - **SCOUT / Flow CoT** performs **vertical recursion**, looping hidden states between internal layers for multi-stage reasoning without emitting intermediate tokens, is fully compatible with standard Hugging Face Trainer workflows, and requires no special data preprocessing  modifications.
>
> ---
>
> ### **3. Efficiency Analysis**
>
> To evaluate inference speed, we measured the wall-clock latency for performing three reasoning iterations.
>
> |Method|Latency (s)|
> |-|:-:|
> |**SCOUT**|0.0327|
> |**CoT-SFT**|0.0448|
>
> - **CoT-SFT:** Three full-model decodings (one forward pass per reasoning step), producing three chain-of-thought tokens.
> - **SCOUT:** Only loops the recursive block three times—without re-running the entire model—hence the lower latency.
>
> SCOUT reduces latency by 27% compared to CoT-SFT (0.0327 s vs. 0.0448 s) for three-step reasoning, demonstrating a clear efficiency advantage.
>
> ---
>
> ### **4. Relationship to Diffusion Models**
>
> Although both Flow CoT and latent diffusion iteratively refine an internal state, they differ in three key ways:
>
> 1. **No Noise Injection**
>   - **Latent diffusion** (e.g. DDPMs) injects noise into the latent representation during training and then learns to denoise over multiple steps.
>   - **Flow CoT** relies entirely on the model’s own predictive capabilities—there is no explicit noise schedule or denoising objective.
>
> 2. **Autoregressive vs. Masked Generation**
> - **Language diffusion** (e.g. LLADA[2]) generates a full token sequence, then in each pass masks out low-confidence tokens and re-predicts them in parallel—explicitly refining token distributions in a “one-shot” update loop.
> - **SCOUT (Flow CoT)** remains fully autoregressive and latent-only: it never re-emits or re-masks tokens during reasoning but instead loops and refines hidden states across layers, then generates exactly one new token per cycle.
>
> ---
>
> ### **5. Statement Discussion**
>
> We first highlight two key ideas:
>
> 1. **Recursive Model Strengthening**: Each pass refines its latent state, so intermediate representations become more accurate and powerful over iterations [3][4][5].
> 2. **Teacher–Student Capacity Matching**: Effective distillation requires teacher sizes aligned with the student’s capacity; oversized teachers early on cause a representational mismatch [6].
>
> When early iterations are supervised by an overly strong teacher, the student’s limited capacity is pushed toward goals it can’t yet model, corrupting its latent trajectory and impairing later loops. For example, R-Distill-EQ sees accuracy fall 38.44→37.17→37.36, whereas SCOUT’s staged teachers (0.5B→1.5B→7B) steadily improve accuracy: 37.44→38.26→39.03. Moreover, relaxing the first-iteration loss weight (Table 3 in [7]) similarly boosts downstream performance, further validating our progressive-distillation approach.
>
> ---
>
> ### **6. Algorithm Details**
>
> The definitions of the head block and tail block are similar to those in [3]. In a typical LLM architecture, we define the first X layers as the head block, the last Y layers as the tail block, and the remaining layers as the recursive block. The specific values of X and Y can either be set manually or determined through ablation experiments to find the most effective configuration. Further details and results from these experiments can be found in Appendix B.1.
>
> Since our framework is compatible with standard frameworks like Hugging Face Trainer, we use the LlamaFactory framework for training, and the lm-evaluation-harness framework for inference. More details about the training and inference processes are provided in Appendix A.
>
> ---
>
> ### **7.  CoQA Performance**
>
> We attribute the slight F1 drop over successive reasoning steps to the CoQA evaluation metric rather than a weakening of SCOUT’s reasoning:
>
> - **Metric sensitivity:** CoQA’s F1 score treats each answer as a bag of characters, so any extra tokens reduce the overlap count—even if the answer remains correct.
> - **Longer outputs:** As shown in Figure 5, iterations 2 and 3 produce answers that are on average 15–20 % longer than iteration 1, introducing more non-overlapping characters and thus lowering the computed F1.
> - **Semantic fidelity:** Manual review shows that later-step answers remain semantically equivalent to the first-step output and often include extra reasoning details, enhancing interpretability.
>
> ---
>
> ### **8. Model Behavior Explanation**
>
> Although GLUE, CoQA, and MBPP are not primarily designed as deep, multi-step reasoning benchmarks—GLUE and CoQA emphasize memory and shallow inference, while MBPP targets code generation—they nonetheless contain non-trivial reasoning components; we therefore include them as relevant, widely used evaluations, and SCOUT still delivers consistent gains by leveraging latent-state recursion to:
>
> - **Memory consolidation:** Prior work shows that stacking LLM layers steadily boosts memory performance—likely because key facts and context get split across loops and reinforced each pass [4]. SCOUT mirrors this effect, improving results on GLUE subsets .
> - **Progressive reasoning:**  Each latent-state loop in SCOUT refines internal representations toward increasingly complex subgoals, boosting final answer quality.
>
> Our results in Table 1, spanning eight benchmarks, show that latent-state recursion consistently improves performance, demonstrating that distributing both memory reinforcement and reasoning refinement across multiple loops benefits a wide spectrum of NLP tasks.
>
> ---
> We hope this clarifies your concerns.
>
> ---
>
> >#### References
> >1. Hao, S. *et al.* “Training Large Language Models to Reason in a Continuous Latent Space.” *arXiv preprint arXiv:2412.06769*, 2024.
> >2. Nie, S. *et al.* “Large Language Diffusion Models.” *arXiv preprint arXiv:2502.09992*, 2025.
> >3. Geiping, J. *et al.* “Scaling Up Test-Time Compute with Latent Reasoning: A Recurrent Depth Approach.” *arXiv preprint arXiv:2502.05171*, 2025.
> >4. Saunshi, N. *et al.* “Reasoning with Latent Thoughts: On the Power of Looped Transformers.” *arXiv preprint arXiv:2502.17416*, 2025.
> >5. Saunshi, N. *et al.* “On the Inductive Bias of Stacking Towards Improving Reasoning.” *arXiv preprint arXiv:2409.19044*, 2024.
> >6. Busbridge, D. *et al.* “Distillation Scaling Laws.” *arXiv preprint arXiv:2502.08606*, 2025.
> >7. Bae, S. *et al.* “Relaxed Recursive Transformers: Effective Parameter Sharing with Layer-Wise LoRA.” *arXiv preprint arXiv:2410.20672*, 2024.

---

> > ### Author Response · Authors · 2025-08-05
> >
> > Dear Reviewer,
> >
> > Thank you very much for taking the time to review our Rebuttal. We sincerely appreciate your valuable comments, which have helped us improve the paper.
> >
> > To ensure the Rebuttal fully addresses all concerns, we would be grateful if you could share any additional concerns you might have. We will make every effort to address them as soon as possible.
> >
> > Best Regards,
> >
> > Authors

---

### Official Review · Reviewer_sQsh · 2025-07-03

**Clarity:** 2
**Significance:** 3
**Originality:** 3
**Rating:** 4
**Confidence:** 4

**Summary:**

The paper proposes Flow CoT and SCOUT, a novel reasoning and lightweight training method.  They propose having recursive reasoning stages within an LLM, where each stage learns from the last stage and improves upon it. Experiments show gains over the prior methods. Moreover, they show qualitatively that each reasoning stage learns how to correctly answer progressively difficult queries.

**Questions:**

Q1: Can you please provide more information on how exactly are the teachers trained, how are they matched for each recursive block?
Q2: How much resources are required

**Ethical Concerns:**

["NO or VERY MINOR ethics concerns only"]

**Limitations:**

yes

**Quality:**

3

**Strengths And Weaknesses:**

Strenthgs:
- Experiments are performed across different reasoning types like commonsense, mathematical, reading comprenhsion, and code generation.
- Results demonstrate improved performancce compared to standard  SFT/distillation and recursive hard label supervision.

Weaknesses:
- Writing could be improved, it is not clear when the input is in latent form and when it is in discrete embedding form
- it is not clear how are teachers trained or matched for each recursive block.
- The paper does not talk about the training or inference resources used
- The paper would benefit from an analysis of accuracy on the entire datasets using each recursive block.

---

> ### Author Rebuttal · Authors · 2025-07-31
>
> **Thank you for recognising the value of our work. Below we detail the additions and clarifications made in the revision, structured around the four points you raised.**
>
> ---
>
> ### **1. Latent Representation vs. Discrete Embedding**
>
> In a standard LLM, the model’s internal hidden states are continuous latent representations that no longer correspond directly to any discrete token in the vocabulary, but instead serve as rich features for deep reasoning.[2]
>
> In SCOUT, the workflow is as follows:
>
> 1. **Input:** discrete tokens (human-readable)
> 2. **Embedding layer:** look up each token’s embedding vector to produce the initial continuous latent representation.
> 3. **Head block & Recursive blocks:**
>    - A **head block** processes latent representation $z^{(0)}$.
>    - Followed by T **recursive blocks**, each transforming the latent state $z^{(t-1)}$ to $z^{(t)}$.
> 4. **Tail block:** processes the final latent state $z^{(T)}$.
> 5. **Output layer (fully connected):** maps $z^{(T)}$ back to discrete tokens (human-readable).
>
>  **Note:** All intermediate representations ($z^{(0)}$, $z^{(1)}$, $...$, $z^{(T)}$) are continuous hidden states that flow only within the model. Only the initial input tokens and the final output tokens are in discrete form. This entire format transformation mirrors the process described in [1], and the use of continuous latent representation to enhance reasoning is motivated by their ability to capture richer information than discrete representations, as noted in [2].
>
> We will add these clarifications in the main text to improve our writing and help readers clearly distinguish between “latent representation” and “discrete embedding.”
>
>
> ### **2. Teacher Model Selection and Matching Logic**
>
> We provided a brief discussion on the teacher model selection and matching in Sections 3.3 and 4.1 of the main text. Below is a more detailed explanation:
>
> **Matching Principle**
> As the number of iterations \(t\) increases, the model’s latent representation capability is enhanced [1][3], so we align each recursion step with a larger-scale/higher-performance teacher model.
>
> **Teacher Configuration**
> To ensure reproducibility, we did not train multiple separate teacher models. Instead, we selected models from the same Qwen2.5 series, which span a broad range of sizes (0.5B–72B) and share a unified vocabulary and knowledge distribution without extra training. We chose the 1.5B, 3B, and 7B variants as our teacher models, applied as follows:
>
> | Teacher Model   | Parameters | Iteration Step \(t\) | Avg Score ↑ | Description                                                                                         |
> |-----------------|------------|----------------------|-------------|-----------------------------------------------------------------------------------------------------|
> | Qwen2.5-1.5B    | 1.5B      | 1                    | 50.84       | Guides the 1st recursion, providing directional guidance without over-constraining.                 |
> | Qwen2.5-3B      | 3B        | 2                    | 55.60       | Guides the 2nd recursion, providing directional guidance without over-constraining.                 |
> | Qwen2.5-7B      | 7B        | 3                    | 60.90       | Guides the 3rd recursion (final output), providing directional guidance without over-constraining. |
>
> **Alternative Strategy When Few Models Are Available**
> If only a single large model is accessible, one can first distill multiple student models of various sizes (e.g., 0.5B, 1B, etc.) and then assign them as teachers for recursion steps 1, 2, and 3 in order from weakest to strongest, preserving the matching principle.
>
> We will expand on the teacher selection and matching logic in more detail in the main text.
>
> ---
>
> ### **3. Compute Resources**
>
> In Appendix A.1 (Training Settings), we provide a comprehensive description of our setup, including:
>
> - **Hardware:** We use a single NVIDIA H20 NVLink GPU (96 GB), a dual-socket Intel® Xeon® Platinum 8457C CPU (20 cores), and 200 GB of RAM. One GPU is sufficient to train each model.
> - **Training configuration:** We employ gradient accumulation to achieve an effective batch size of 128. Training is performed in bfloat16 (bf16) precision.
> - **Optimization hyperparameters:** Learning rates, optimizer settings, and additional details are listed in Appendix A.1.
>
>
>
> We will add a cross-reference in the main text’s experimental settings section to direct readers to Appendix A.1.
>
>
> ---
> ### **4. Dataset-wide Performance Analysis Across Recursive Blocks**
>
> In Table 1 of the main text, the second column “Iter” corresponds to the average performance after each recursive block:
>
> | Method | Iter | Avg Score |
> | ------ | ---- | ------------ |
> | SCOUT  | 1    | 37.44%       |
> |        | 2    | 38.26%       |
> |        | 3    | 39.03%       |
>
> - The results demonstrate a monotonic increase in performance as the number of iterations grows, validating the effectiveness of progressive distillation. We have discussed this trend in detail in Section 4.2.
> - To present this more clearly, we will add an annotation in Table 1 clarifying that “Iter represents the number of recursive blocks.”
>
>
>
>
> ---
>
> Thank you once again for your valuable feedback.
>
> ---
>
> >#### References
> >[1] J. Geiping *et al.* “Scaling up test-time compute with latent reasoning: A recurrent depth approach.” *arXiv preprint arXiv:2502.05171*, 2025.\
> >[2] S. Hao *et al.* “Training large language models to reason in a continuous latent space.” *arXiv preprint arXiv:2412.06769*, 2024.\
> >[3] N. Saunshi *et al.* “Reasoning with latent thoughts: On the power of looped transformers.” *arXiv preprint arXiv:2502.17416*, 2025.

---

> > ### Author Response · Authors · 2025-08-05
> >
> > Dear Reviewer,
> >
> > Thank you for carefully reading our rebuttal and for the constructive feedback you provided on our work.
> > We hope the additional details about latent versus discrete representations, teacher selection, compute resources, and dataset-wide results have addressed your concerns. If you have any further questions or suggestions, please let us know—we will be happy to clarify them promptly.
> >
> > Best regards,
> > Authors

---

### Author Response · Authors · 2025-08-09
**General Response**

We sincerely thank the Reviewers, ACs, SACs, and PCs for their time and thoughtful evaluation. We appreciate the recognition of our paper’s **novelty** (rHmT, Z32A), the **breadth of the evaluation** (sQsh, rHmT), and the **effectiveness** of the approach (sQsh, Z32A).

During the rebuttal and discussion phase, we endeavored to address all points raised by the reviewers and focused our updates on two fronts: (1) **additional experiments**—including  new results demonstrating scalability and an efficiency analysis—and (2) **clarifications**—including teacher model selection and matching, the relationship to diffusion-style methods, and the rationale behind our backbone choice. We will incorporate these updates, along with related editorial improvements, into the final version.

In brief, our contributions mainly include two aspects: we propose **Flow CoT**, which conceptualizes recursive reasoning as a **progressive, depth-aware (structured) cognitive evolution**; and we introduce **SCOUT**, a lightweight fine-tuning framework that instills Flow CoT capabilities in pretrained LLMs via **progressive distillation (with staged teachers)**, enabling recursive reasoning without additional pretraining.

We believe that Flow CoT will offer the reasoning community a clear, practical perspective and contribute to its development. We also hope that SCOUT will inspire further research and help advance the community.

Once again, we are grateful to the Reviewers, ACs, SACs, and PCs for their efforts and constructive feedback that strengthened the manuscript. Thank you for your time and consideration.

---

### Decision · Program_Chairs · 2025-09-17

**Decision:**

Accept (poster)

**Comment:**

This paper proposed a framework called SCOUT for reasoning model fine-tuning where a progressive distillation method and a retrospective reasoning module are adopted. In the progressive distillation, the authors use a series of QWen models with different model sizes as teacher models to provide supervise signals for the intermediate steps in a recursive process. At the same time, the retrospective module consists of both self-attention and cross-attention components to incorporate the original latent information and the intermediate reasoning results.

Reviewers agreed that the proposed methodology is novel and the experimental evaluation is well designed.
However, some of them also pointed out the paper is not well written and need to be improved. Using a series of QWen models as teachers to train a small model to improve its reasoning capability. It seems that such a method is not scalable in practice. In addition, this paper only uses Qwen models in the experiments, and the conclusion might not be able to general applicable to other model families because there are papers pointing out that some strange behavior of Qwen models.